# Continuous Evaluation of Shear Wave Velocity from Bender Elements during Monotonic Triaxial Loading

**DOI:** 10.3390/ma16020766

**Published:** 2023-01-12

**Authors:** Ahmed Khalil, Zahid Khan, Mousa Attom, Kazi Fattah, Tarig Ali, Maruf Mortula

**Affiliations:** Department of Civil Engineering, American University of Sharjah, Sharjah P.O. Box 26666, United Arab Emirates

**Keywords:** shear wave velocity, bender elements, triaxial testing, micromechanics

## Abstract

Few researchers have attempted to experimentally evaluate the low-strain shear wave velocity (*Vs*) of specimens undergoing large strain deformations. They report that the *Vs* is practically unaffected by the strains, and the reasons behind this behavior are not fully understood. This study presents the continuous measurement of low-strain *Vs* with bender elements (BE) during monotonic shearing of two sand specimens in a triaxial device. The results are analyzed using a micro-mechanical model based on contact theory. The results of this study confirm that the *Vs* values from BE measurements are unaffected by an increase in axial strains that are induced by a separate mechanism. The micro-mechanical model predictions of *Vs* agree well with the results of this study and with the results of previous studies. They show that the mean effective stress and increase in inter-particle stiffness controls the low-strain stiffness despite a global increase in strains during monotonic loading.

## 1. Introduction

Dynamic properties, such as shear wave velocity (*Vs*) and damping ratios (*D*), as a function of wide range of shear strains are required for the dynamic analysis of structures and construction sites. Low-strain *Vs* is required to classify sites for seismic analysis, and corresponding amplification factors for ground accelerations are determined. Dynamic properties are typically obtained by conducting field and laboratory tests (Khan et al. [1]; Khan et al. [2]; Khalil et al. [3]; Irfan et al. [4]). These tests have various limitations and testing biases related to boundary conditions, assumptions, attainable strains and frequencies (e.g., Khan et al. [1]; Clayton et al. [5]). A combination of tests is required to evaluate the dynamic properties at the required range of strain levels; however, the results seldom reconcile. Researchers and practitioners need a single test method that would provide the dynamic properties at the required range of strain levels instead of resorting to a combination of test methods.

In one approach, a few researchers have attempted to perform continuous BE tests during the shearing of a specimen during triaxial testing (Chaney et al. [6] and Ismail and Rammah [7], Dutta et al. [8]; Styler and Howie [9]). The outcomes of the studies are contrary to expectations, which caused more interest; the main reasons for the unexpected results are still not fully understood, but should be in order to better design future experiments. Typically, *Vs* as a function of shear strains is measured with a combination of resonant column (RC) and cyclic triaxial (CT) devices which require additional resources (e.g., Khan et al. [1], Khalil et al. [3]). Bender elements (BE) have been extensively used to determine low-strain *Vs* (Dyvik and Madshus [10], Shirley and Hampton [11], Viggiani and Atkinson [12], Kuwano and Jardine [13], Gu et al. [14], Khalil et al. [3]; Irfan et al. [4]). *Vs* measurements with BE are carried out at very low strains because the elements in BE systems are not capable of producing large strains in the specimens. Researchers have always experimented using BE or other high frequency measurements such as ultrasonic testing (UT) at large strains without conducting dynamic CT or RC tests to complement the results.

~Previous attempts to use BE and UT during monotonic shearing of specimens in a triaxial device show inconclusive results (Dutta et al. [8]; Styler and Howie [9]). These studies report that the low-strain stiffness (*Vs* measurements from BE or UT) is unaffected by the strains that are generated by shearing the specimen in a triaxial device. The initial increase and subsequent decrease in *Vs* measurements is attributed to the contractive and dilative stages of the specimen, respectively. The observed behavior of the *Vs* from BE tests is mostly explained in qualitative terms; therefore, more testing and analysis is required to better understand the behavior. Micro-mechanical analysis can provide important insights that can help us to understand the mechanisms that control the low-strain stiffness. It can provide directions that can lead to the development of a single test method that can evaluate *Vs* at the required range of strain levels.

This study presents the results of experimental program that includes the evaluation of *Vs* from CT tests as a function of shear strain and low-strain *Vs* from continuous BE measurements during the monotonic loading of specimens in a triaxial device. The *Vs* values are also computed from the stress–strain curves of the monotonic loading of specimens. All tests are performed at two different confinements and for two identical sand specimens for each test, to ensure repeatability. The sands are characterized with scanning electron microscope (SEM), energy-dispersive X-ray spectroscopy (EDS) and confocal microscopy to better understand the inter-particle interaction. The BE results are further analyzed with Hertz-Mindlin contact theory (Hertz [15]; Mindlin [16]) to study the evolution of *Vs* from the perspective of interparticle forces, mean effective stresses and change in void ratio.

## 2. Background and Literature

The bender element (BE) test induces high frequency (>1 kHz) stress waves in the specimen (e.g., Irfan et al. [4]; ASTM D8295-19 [17]). The strains generated in the specimen are typically low to very low; therefore, a combination of other tests, such as RC and CT, are used to evaluate the dynamic properties at other required strain values. In the BE test, a transmitter generates a stress wave with a short duration impulse of step or sine function. The duration of the pulse determines the frequency content of the propagating wave. The receiver located at a known distance detects the arrival, and velocity is obtained by simply dividing the known distance by the time of propagation. Velocity analysis can also be performed in the frequency domain with phase velocities.

The triaxial test is an ASTM standard (ASTM D7181-20 [18]) in which monotonic axial loads are applied to the specimen until failure, and a stress–strain curve is obtained for further analysis. The stress–strain curve can be obtained from monotonic triaxial tests under different confining and drainage conditions. Subsequent analyses of the curve provide elastic and secant moduli at different axial strain levels. Changes in volumetric strains as a function of axial strain represent the changes in void ratio during loading.

In cyclic triaxial (CT) tests (Khalil et al. [1]; Khan et al. [19]), dynamic axial loads are applied to the specimen to evaluate dynamic properties as a function of shear strains at different confining and drainage conditions. The operable frequencies in CT tests range from less than 1 Hz to 10 Hz; however, only medium to large strains can be induced in the specimen. Low strain tests such as bender elements (BE) are performed to compliment the strain range.

The resonant column (RC) test is an ASTM standard (ASTM D4015-21 [20]) in which dynamic torsional loads are applied to evaluate the dynamic properties of soils at low to medium strain levels. The RC test has different testing biases which should be considered to correct the results (Khan et al. [1]; Clayton et al. [5]; Khan et al. [21]). The results become increasingly biased as the stiffness of the specimen increases, due to contributions from the base of RC device. Salient features of the above tests, such as the operating range of frequencies and induced strain levels, are presented in Table 1. The advantages and limitations that can affect their results are also presented. The frequencies and strains that are generated in typical geotechnical problems involving dynamic loads, such as machine vibrations and earthquakes, are presented for comparison purposes.

A hyperbolic model is often used to curve fit the variation of shear wave velocity with shear strains (e.g., Khalil et al. [3], Khalil [22]; Hardin and Drnevich [23]). Site response analysis requires the hyperbolic model for each soil layer of the construction site as an input for analysis. Equation (1) presents the functional form of the modified hyperbolic model.
(1)$$\frac{G}{G_{max}}=\frac{1}{1+\gamma_{h}}$$
where *γ_h_* is the hyperbolic strain which is defined by Equation (2). *G_max_* is the shear modulus measured at the lowest possible strain level and *G* is the shear modulus corresponding to higher strain levels.
(2)$$\gamma_{h}=\frac{\gamma }{\gamma_{h}}\;\;\left(1+a\exp\;(-b\frac{\gamma }{\gamma_{r}}\right))$$

Model parameters *a* and *b* are defined in terms of curvature coefficients and *γ_r_* is the reference strain, indicating the beginning of non-linear behavior. These parameters are obtained from the non-linear curve fitting of the model to measured data.

The shear wave velocity of soils depends on many test variables, such as confinement, packing density (void ratio), particle shape and roughness and soil type. The effect of void ratio is significant if all other variables are kept constant. The packing density (void ratio) determines the structure of soil matrix and number of interparticle contacts (*C_n_*). The coordination number (*C_n_*) is the number of contacts that a soil particle will create with other surrounding particles in a packing. Sand particles are arranged randomly, and their *C_n_* is considered to be a function of void ratio of the soil matrix; however, this is not always true. Many studies have shown that *C_n_* can be related to void ratio; however, any changes in which rearrangement of particles occurs can change the void ratio without significantly affecting the *C_n_* (Walton [24]). A summary of selected studies that relate *C_n_* to the void ratio of specimen are presented in Table 2.

Chang et al. [29] experimentally determined soil stiffness at small strains by establishing a relationship between *C_n_* and e. Magnanimo et al. [30] showed that random samples prepared with different preparation methods had similar e, but different *C_n_* which shows the sensitivity of *C_n_*-e to sample preparation. Smith et al. [25] investigated the *C_n_*-e relationship for random packing by comparing experimental data with regular arrays, including face-centered cubic (FCC), hexagonal close-packed (HCP) and simple cubic (SC) arrays. However, there is no agreement between the *C_n_*-e data points for body-centered cubic (BCC) array. Ouchiyama & Tanaka [26] and Arakawa & Nishino [31] presented theoretical expressions for the *C_n_*-e relationship which are in good agreement with the expression derived by Smith et al. [25]. Furthermore, Suzuki et al. [32] incorporated a Gaussian error function to the model proposed by Tory et al. [33] which shows good agreement with the experimental results of Bernal and Mason [34] and Gotoh [35]. German et al. [28] presented correlations between *C_n_* and e for various packing types, including ordered, dispersive, random, partially densified and fully densified packing.

### 2.1. Micro-Mechanical Evaluation of Internal Forces and Vs

Contact models for the random packing of spheres have been developed by researchers for evaluation of the Poisson’s ratio of the packing (Hertz [15], Mindlin [16], Digby [36], Walton [24], and Norris and Johnson [37]). Duffaut et al. [38] presented an expression for the effective shear modulus combining Digby’s result (Digby [36]) with Mindlin’s extended solution (Mindlin [16]) that includes frictional tangential contact stiffness. The normal contact force (average) for an isotropic random assembly of particles is estimated with the micromechanical formulation proposed by Rothenburg and Bathurst [39] for a given void ratio e, coordination number *C_n_* and isotropic confinement σ_o_ by
(3)$$f_{n}=\frac{4\pi \left(1+e\right)R^{2}\sigma_{0}}{C_{n}}$$
where *C_n_* is the coordination number that can be computed from several empirical equations presented in Table 2. During monotonic triaxial loading, the specimen’s void ratio decreases during the compression stage until the beginning of dilation. This phenomenon is represented by changes in volumetric strains. The *C_n_* number can be assumed to remain practically constant during the compression stage (negative volumetric strain); therefore, the normal contact force (*f_n_*) can be considered a function of void ratio and mean effective isotropic confinement. The change in radius R of the particles is negligible due to their large elastic moduli compared with the moduli of the packings.

If two elastic particles are in contact under the action of a normal force *f_n_*, the Hertz theory (Hertz [15]) relates the contact area with the radius and the normal displacement between the particles. Duffaut et al. [38] extended this inter-particle mechanism and presented the final expression for bulk modulus of the random assembly of particles (Equation (4)).
(4)$$K_{\mathrm{dry}}={\left(\frac{C_{n}^{2}{\left(1-\phi \right)}^{2}G_{g}{}^{2}\sigma }{18\;\pi^{2}{\left(1-v\right)}^{2}}\;\right)}^{\frac{1}{3}}$$
where *ϕ* is porosity (related to void ratio), *G_g_* is shear modulus of the particles, σ is the mean effective stress, and *ν* is the Poisson’s ratio of the particles. The conversion of bulk modulus to shear modulus is based on friction between the particles that will either allow no slippage or complete slippage under tangential force at the inter-particle contact. The relationship between bulk and shear modulus is provided in Equation (5).
(5)$$G_{\mathrm{dry}}=\frac{3}{5}\left[1+\frac{3\left(1-v\right)}{2-v}f\left(\mu \right)\right]K_{\mathrm{dry}}$$

The term f(*μ*) ranges from 0, when there is complete loss of inter-particle contact, to 1, when no slippage occurs. f(*μ*) represents the Mindlin friction term, which essentially is a function of tangential force, normal force, and the coefficient of friction at inter-particle contact. f(*μ*) can be estimated in various ways; however, Equation (6) presents a simpler approach of estimating f(*μ*) for a given Poisson’s ratio of the particles (*ν*) and of the matrix (*ν _dry_*).
(6)$$\left(\mu \right)=\frac{1}{2}\left(\frac{2-v}{1-v}\right)\left(\frac{1-4\;v_{dry}}{1+v_{dry}}\right)$$

Particles rearrange and deform when loaded, and the degree of disturbance depends on the induced strain levels. Noticeable particle rearrangement and changes in fabric occur at medium to large strains (*γ* > 10^−3^). The degree of changes depends on many factors, such as surface roughness, gradation of particles and forces at the inter-particle contacts. In the beginning of deviatoric loading in triaxial tests, the main mechanism is dominated by particle deformation with slight slippage that can also cause crushing of asperities of the surface. Particle rearrangement occurs when strains exceed 10^−3^, and the degree of rearrangement depends on the coordination number and inter-particle forces, especially *f_n_* (Equation (3)). Complete slippage of the contact area at the particle is assumed during the dilatational stage, in which the void ratio increases and *C_n_* decreases. At very low strain measurements, such as in BE, particle deformation that depends primarily on the elastic properties of the sand particles controls the stiffness.

Equation (3) suggests that during the initial stages and even within the contraction stage, the low strain stiffness (if measured independently) shall increase due to an increase in effective stress at contacts. The mean effective stress at the inter-particle contacts increases with an increase in deviatoric stress, and primarily governs the stiffness in addition to the effect of change in void ratio. On the other hand, velocity is expected to decrease during dilatational stage.

### 2.2. Continuous Vs Measurements during Monotonic Triaxial Loading

Few researchers have attempted to evaluate the low-strain stiffness of specimens that are sheared monotonically in triaxial testing. Chaudhary et al. [40] presented the results of *Vs* measurements of the Toyoura sand with BE tests at a constant effective stress ratio. The results show that the shear wave velocity practically remains constant during the contraction stage; however, *Vs* starts to decrease following the phase transformation of the specimen from contraction to dilation.

Styler and Howie [9] evaluated the variation of low-strain *Vs* with the change in void ratio (*e*) of the specimens during triaxial loading. Fraser river sand is tested at different confinements; however, the stress ratio is kept constant at 2.0. The results show that the *Vs* increases as the void ratio decreases during the contraction stage. During phase transformation and throughout the dilation stage, the *Vs* steadily decreases as the void ratio increases. They show that the rate of increase during contraction stage is not the same as the rate of decrease in *Vs* during the dilation stage.

Dutta et al. [8] presents continuous evaluation of low-strain *Vs* with disk-shaped planar transducers during triaxial testing. The variation of *Vs* is presented as a function of axial strains (*ε_a_*). The study presents qualitative discussions and argues that the mean *C_n_* controls the evolution of *Vs* during the contraction and dilative stages. The results also confirm the previous findings that the *Vs* increases during the contraction stage and then starts to decrease after phase transformation to the dilative stage.

All studies indicate that low-strain *Vs* is not affected by the level of shear strain in the specimen and the observed changes in *Vs* are attributed to the changes in void ratio (or *C_n_*) during monotonic axial loading of the specimens. The change in void ratio is relatively simple to calculate from the change in volumetric strains; however, there is no experimental evidence that *C_n_* changes proportionally to void ratio. Table 2 presents different expressions that relate void ratio to *C_n_*. The functional form of the expressions ranges from linear to exponential which indicates the level of uncertainty in relating *C_n_* to *e,* especially for a random assembly of irregular particles.

## 3. Materials Properties

Sand samples are collected from a borehole located in Al Shamkhah area of Abu Dhabi, United Arab Emirates (UAE). A typical cylindrical specimen of soil has a diameter of 7 cm and a length of 14 cm in the study. Various routine tests are performed, such as moisture content, index density and gradation analysis. Table 3 presents the main properties of the sands and samples. Figure 1 presents the particle size distribution curves of the two sand samples.

The sands are also characterized with scanning electron microscopy (SEM), energy-dispersive X-ray spectroscopy (EDS) and confocal microscopy (CFM). SEMs produce images that are magnified by using electrons instead of light, whereas EDS detectors separate the characteristic x-rays of different elements into an energy spectrum. In addition, EDS provides chemical composition of the material and creates element composition maps over a larger raster area using the energy spectrum.

SEM and CFM are used to visualize the grain structure, grain size and surface roughness. Sample A has angular-shape particles, whereas sample B has rounded to sub-rounded particles (Figure 2). Visual inspection of these figures also provides an estimate of probable *C_n_* which is anticipated to be more than 10 in 3D space. Furthermore, three spectra each are obtained for both materials with EDS to determine the chemical compositions (Figure 3). The results of three spectra are averaged for each material. Both materials exhibit quite similar chemical compositions with slight variations in the Calcium and Silicon content.

Table 4 summarizes the averaged chemical compositions of both samples. Silicon dioxide (SiO_2_) was the dominant chemical composition of both samples A and B. Both samples show inclusion of sodium chloride because of their proximity to the Arabian gulf, and that the primary sediments at depth originate from the deep sea carbonate platform (Khan et al. [2]). Moreover, both materials contain calcium inosilicate mineral (CaSiO_3_) commonly known as wollastonite. Wollastonite primarily contains calcium oxide (CaO), an ingredient used in Portland cement. Presence of minerals other than SiO_2_ in relative abundance is indicative of heterogeneous composition and can be one of the possible reasons for differing elastic deformation characteristics compared with pure silica sand.

Representative two-dimensional topographies for both samples are obtained using confocal microscopy through a surface area of approximately 550 μm^2^. Figure 4 presents the surface profile of typical particles from sample A and B. The profile indicates rough surfaces with asperities that are brittle and can break upon deformation. The inter-particle slippage is predicted to be smaller due to increased inter-particle friction caused by asperities.

## 4. Experimental Program

Dynamic triaxial testing (ASTM D3999-91 [41]) and monotonic triaxial loading (ML) are performed with a dual mode cyclic triaxial test apparatus supplied by VJ Tech (UK). CT equipment also includes BE fixed to the top and bottom platens with independent controls. All tests are conducted under two confining pressures (*σ*_3_) of 150 kPa and 300 kPa. Identical specimens with similar void ratios are produced for CT and ML tests from the two types of sands. Continuous BE measurements are taken during ML tests at regular increments of axial strains. Low strain measurements with BE are also taken before the start of CT tests for comparison with BE measurements performed before the start of ML tests on the other identical specimen.

An input signal of 1 Hz with 10 cycles is used for CT tests. The load and displacement time signals are fitted with a simulated signal to create symmetrical loops for further analysis following standard procedure (Khalil et al. [3], Khalil et al. [22]; Kumar et al. [42]). Measured and fitted signals for the cyclic triaxial data are shown in Figure 5a,b. Figure 5c shows the comparison of hysteresis loops from the measured and fitted signals. For clarity, the load and displacement amplitudes are normalized in the figures. Elastic modulus (*E*) is computed at various strain levels and subsequently converted to *G* and *Vs*.

A baseline measurement of low-strain *Vs* with BE is obtained at axial strain (*ε_a_*) of 0% before conducting monotonic triaxial loading (ML). The soil specimen is then subjected to monotonic triaxial loading (shearing at regular path) with simultaneous BE tests at regular intervals. An axial strain rate of 1 mm/min is adopted for the ML testing. The BE measurements are performed at a frequency of 5 kHz, and 10 signals are stacked to improve the signal-to-noise ratio. *Vs* is estimated in the time domain after dividing the distance (between the elements) by the propagation time of the wave front. The distance between the elements is dynamically adjusted for different axial strains (*ε_a_*). Secant modulus (*E_sec_*) at predetermined axial strains is computed from stress–strain curves. The secant elastic modulus (*E_sec_*) is then converted to secant shear modulus (*G_sec_*), and then finally to *Vs*. Uncemented to lightly cemented sands can vary in Poisson’s ratio from 0.15 to 0.3 (Santamrina et al. [43]), therefore a value of 0.25 is chosen for conversion between moduli.

Typical time histories and their respective Fourier spectra recorded at the receiver (Rx) for two randomly selected shear strains (7% and 14%) are shown in Figure 6. To avoid near-field effects, at least one wavelength was ensured between the transmitter and receiver in this study (Arroyo [44], Lee [45], Khalil et al. [3]; Khalil [22]). The peak power spectrum in BE tests of a wave front propagating at an average *Vs* of 300 m/s is centered around 4500 Hz (Figure 6). The average wavelength in this study is 6.6 cm compared with the propagation distance of 14 cm, which ensures at least on wavelength between the transmitter (Tx) and receiver (Rx).

## 5. Results and Discussion

Figure 7a,b presents the relationship between deviatoric stress (*q*) and axial strain (*ε_a_*) for Samples A and B from ML testing. The stress–strain curves are compared with curves presented by Dutta et al. [8] at two confinements. The results from Dutta et al. [8] are presented for a predominately siliceous sand at 100 kPa. The behavior of sands in present study is noticeably different from Dutta et al. [8]. There are many possible reasons for the difference, such as different void ratios, particle shape and stress ratios, and difference in mineral composition of the sand particles. The ductile behavior of sand specimens (A and B) is evidenced by yielding at significantly larger axial strains. Sample A at 150 kPa deforms almost linearly with strain despite having a relatively smaller void ratio (e = 0.81) compared with Sample B (e = 0.99). Figure 7 also shows stick-slips (small serrations on the stress–strain curve) which has been reported by other researchers (Nasuno et al. [46]). The stick-slips are typically observed in particles with rough to very rough surfaces.

Figure 8 presents the variation of normalized values of *Vs* with axial strain (*ε_a_*) from continuous BE tests, CT tests, and ML tests. All *Vs* values are normalized to *Vs* value BE tests that correspond to an axial strain of 0%. The value of 0% is replaced with a very small value of 10^−6^ for plotting on logarithmic scale. Hyperbolic model (Equation (1)) is fitted to cyclic triaxial data and then extended to *Vs* values computed from the stress–strain curves from ML tests. The *Vs* values from CT and those from ML decrease with increase in strain level. It is interesting to note that the extended hyperbolic model fits the *Vs* values from both CT and ML despite different mechanisms of loading, except for sample A at 150 kPa. ML is not a dynamic test; however, CT tests also do not operate at high enough frequencies to involve the inertial response of a single degree of freedom (SDOF) system. RC tests operate at much higher frequencies; therefore, the hyperbolic model fitted to RC data typically does not reliably predict CT results.

Figure 8 shows that the low-strain *Vs* values from BE tests slightly increase with increase in axial strain and seem to be unaffected by the increasing axial strain in the specimen. The scatter in the *Vs* from BE tests at very large strains is due to lower quality signals, possibly because of the distortion and misalignment of the transmitter and receiver. Figure 8 indicates that the behavior of low-strain stiffness, representing small scale deformations of the specimen, is different from the evolution of stiffness corresponding to larger-scale deformations. Previous studies have also noticed a slight increase in low-strain *Vs* during the contraction stage of the sample during the ML test (Dutta et al. [8]; Styler and Howie [9]). They attribute this slight increase in low-strain *Vs* to a decrease in void ratio anecdotally; however, relating the behavior to an observed change in void ratio is not sufficient to understand its cause.

The low-strain *Vs* of soils depends on the initial void ratio to some extent; however, as noted earlier, the mean effective stress governs the magnitude of normal force at the interparticle contacts (Equation (3)). The shear stresses caused by the propagation of shear waves produce tangential forces at the inter-particle contacts, and the deformation response at the contacts to tangential forces depends on the magnitude of the normal force (*f_n_*) at the contacts (Duffaut et al. [38]). The stiffness and therefore low-strain *Vs* are expected to increase with an increase in normal forces. The increase in axial stress (*σ_a_*) during ML tests results in an increase in mean effective stress in the specimen. A micro-mechanical model based on contact theory (Equation (5)) is used to theoretically evaluate the low-strain *Vs* for samples A and B. The Mindlin friction term f (μ) is computed from Equation (6) and assumed to remain practically constant during the contraction stage. The prediction of *Vs* (Equation (5)) at different stages of the sample deformation (axial strains) is also presented in Figure 8.

The micro-mechanical prediction of *Vs* indicates a slight increase with an increase in the mean effective stress. Micro-mechanical prediction of *Vs* does consider the change in void ratio; however, the decrease in void ratio is not the main cause of slight increase in *Vs.* Equation (3) suggests that the decrease in void ratio should cause a decrease in normal force at the inter-particle contacts if the mean effective stress remains unchanged. The contact theory assumes that a decrease in void ratio will cause an increase in the number of contacts (not always true) which will result in many but relatively smaller contact forces. The overall stiffness therefore remains unaffected. Figure 8 predicts an increase in low-strain *Vs* of about 10% to 15%, which is comparable to the increase measured in BE tests. The dilation of tested specimens occurs very late in the loading cycle; therefore, the effect of an increase in void ratio (decrease in inter-particle stiffness) on the low-strain *Vs* is not clear.

The comparison of the measured data from this study and two other studies (Dutta et al. [8]; Styler and Howie [9]) from the literature are presented in Figure 9. The compared studies include tests that are performed at confinement of 100 kPa which are compared to the results of this study at their closest confinement of 150 kPa. The results from CT tests are not compared because the selected studies did not perform the CT tests. The logarithmic scale is decreased to better visualize the comparisons. The normalized variation of low-strain *Vs* from Dutta et al. [8]; Styler and Howie [9] agree well with the results from this study. The normalized *Vs* that are converted from secant shear modulus (stress–strain curves) of Dutta et al. [8] are slightly higher than this study; however, the trend is almost similar. The results from Dutta et al. [8] shows that the low-strain *Vs* values (BE testing) decrease after an initial increase. They attributed the initial increase and subsequent decrease in low-strain *Vs* to the contraction and dilatational phase, respectively.

The behavior of random packing of spheres such as in sands can be predicted well with contact theories based on the work of Hertz and Mindlin. The accuracy of contact theories still needs to be investigated for cohesive soils such as clays and some silts. Other theoretical models involving the response of structural materials and systems to complex contact and non-contact loads, along with volumetric damage models, can be developed to understand the behavior better (Sosnovskii et al. [47], Shcherbakov [48]).

## 6. Conclusions

In this study, the variation of shear wave velocity (*Vs*) with axial strain (*ε_a_*) during monotonic triaxial compression is obtained and compared with the literature. The *Vs* is measured from BE tests, CT tests, and stress–strain curves. The BE tests are continuously performed at regular intervals during the monotonic axial loading of the specimen. The *Vs* from stress–strain curves is calculated after converting *E_sec_* to *G_sec_*. The CT testing and ML tests along with BE tests are performed on two identical specimens from each sand type. The behavior of low-strain *Vs* from BE is analyzed by using contact theory. The main conclusions of the study are presented in the following.

The frequency content of the BE signals tends to shift towards higher frequencies with an increase in axial strain (*ε_a_*), possibly because of larger average stress at the location of bender elements.The stress–strain behavior of tested sands (heterogeneous mineral composition) is markedly different from siliceous sand; this can be attributed to many factors, such as density of the packing, particle orientation and degree of freedom, and particle shape and roughness.The *Vs* from BE is unaffected by the strains imposed during monotonic loading. This behavior needs further numerical and experimental investigations to understand and decouple the complex interaction of average inter-particle stress and reduction in void ratio (*e*) during monotonic loading.The contact theory predicts the variation of low-strain *Vs* obtained from BE tests well.The *Vs* values calculated from the stress–strain curves and the *Vs* values from CT follows the hyperbolic model with one exception.The low-strain *Vs* results of this study compare well with the *Vs* results of past studies. The trends in *Vs* values calculated from the stress–strain curves are also comparable.The micro-mechanical models are based on contact theory of two spheres in contact. The models may not be valid for cohesive soils, such as clays and some silts. Additional parametric tests based on a similar approach on clayey soils are needed to understand their behavior.

## Figures and Tables

**Figure 1 materials-16-00766-f001:**
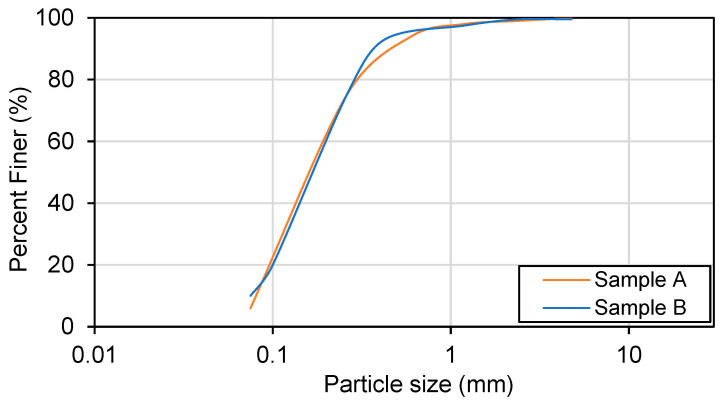
Particle size gradation of sands.

**Figure 2 materials-16-00766-f002:**
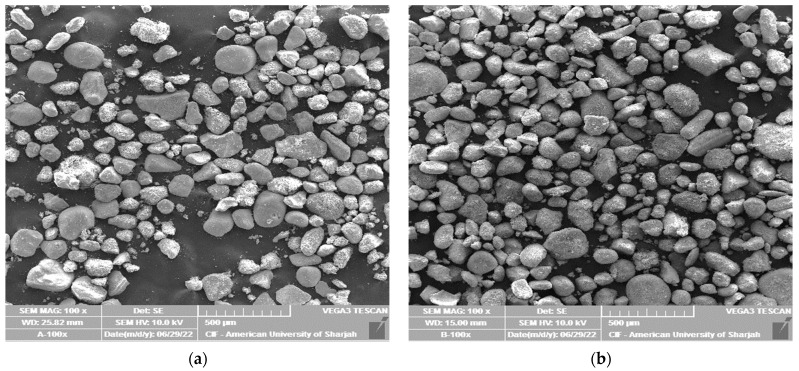
SEM images with a magnification of 100× of tested soil materials; (**a**) sample A and (**b**) sample B.

**Figure 3 materials-16-00766-f003:**
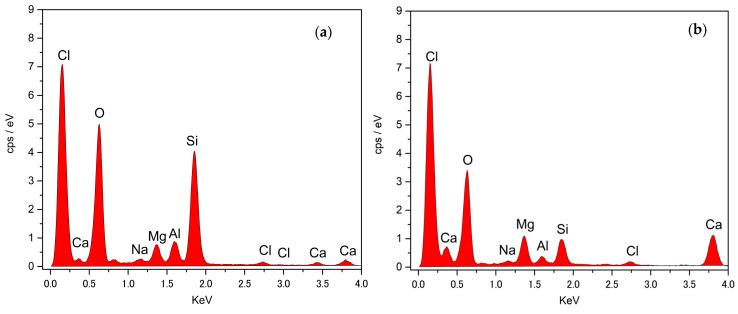
Typical EDS spectra of (**a**) Sample A, and (**b**) Sample B.

**Figure 4 materials-16-00766-f004:**
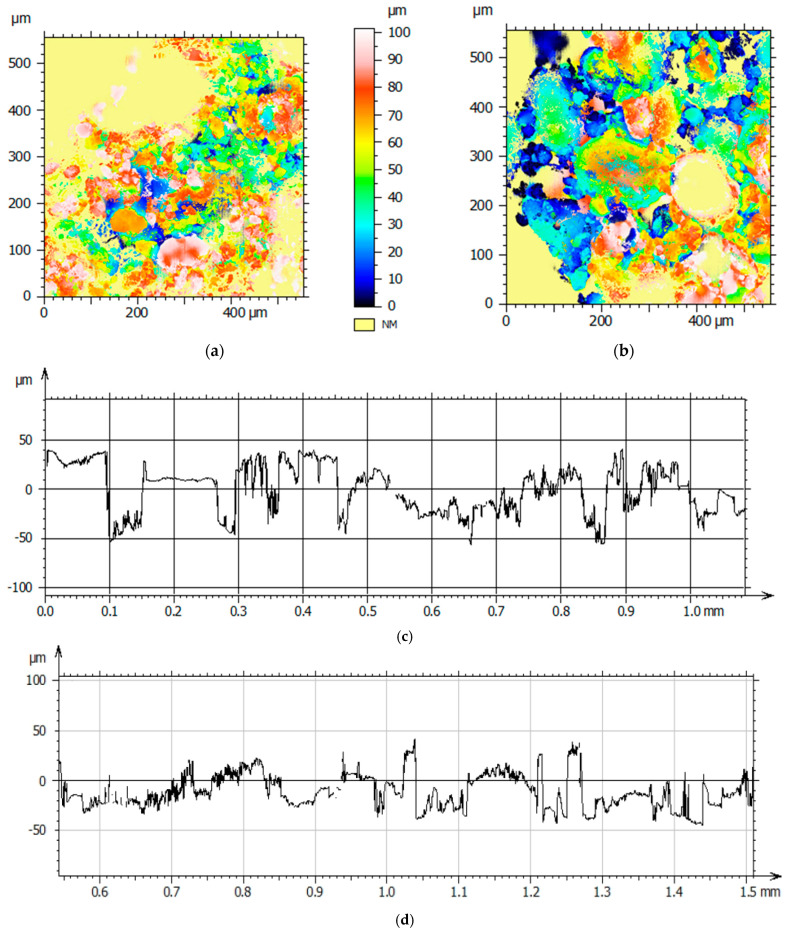
Scanned surfaces of (**a**) sample A and (**b**) sample B using confocal microscopy and 2D surface topographies of (**c**) sample A and (**d**) sample B.

**Figure 5 materials-16-00766-f005:**
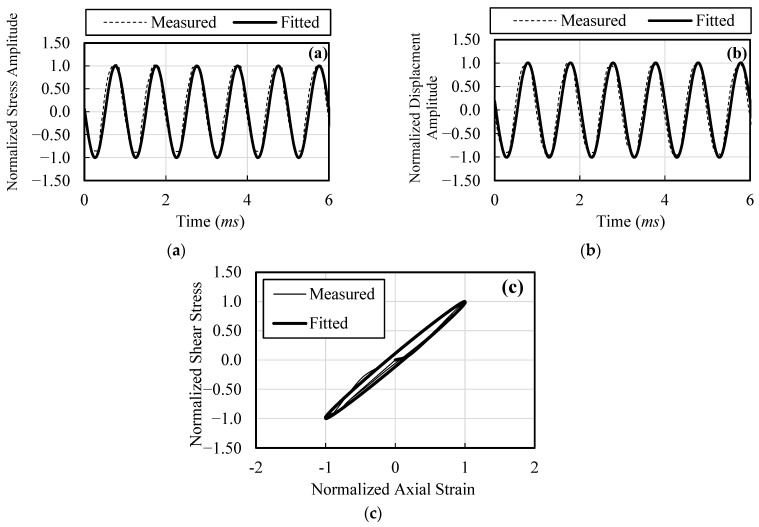
Typical matching of normalized signals and loops from CT tests. (**a**) Stress amplitude; (**b**) Displacement amplitude; (**c**) Hysteretic loops.

**Figure 6 materials-16-00766-f006:**
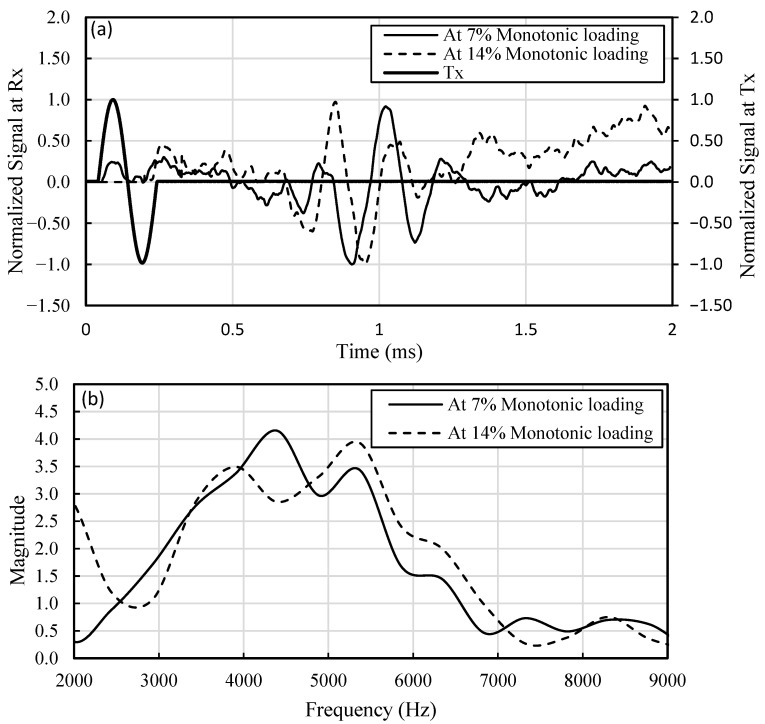
(**a**) Time histories at the bender element (BE) receiver (Rx), and (**b**) Fourier spectra of receiver signals. Tx corresponds to the transmitter (trigger) signal.

**Figure 7 materials-16-00766-f007:**
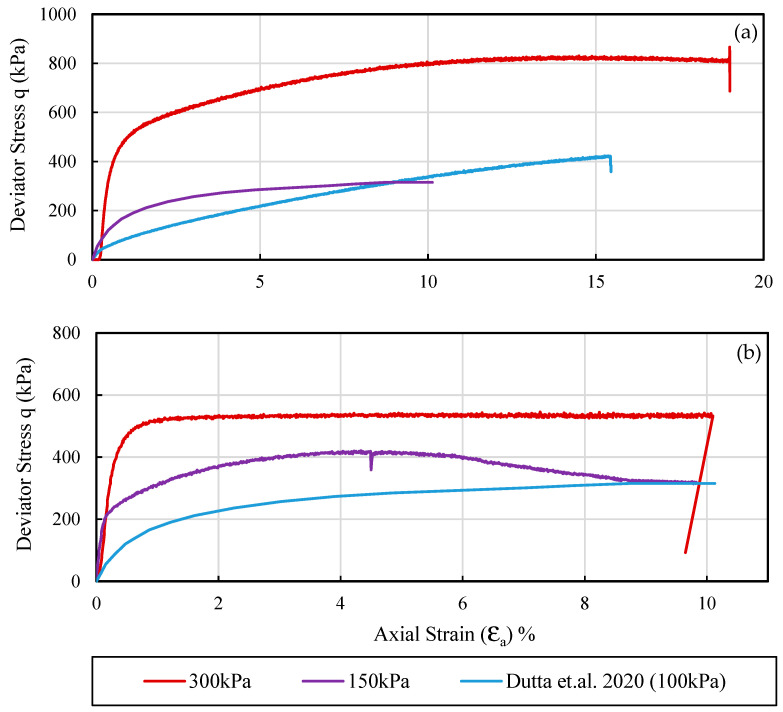
Stress–strain curve from siliceous sand (Dutta et al. [8]) at σ_3_ of 100 kPa and this study (**a**) sample A, and (**b**) sample B.

**Figure 8 materials-16-00766-f008:**
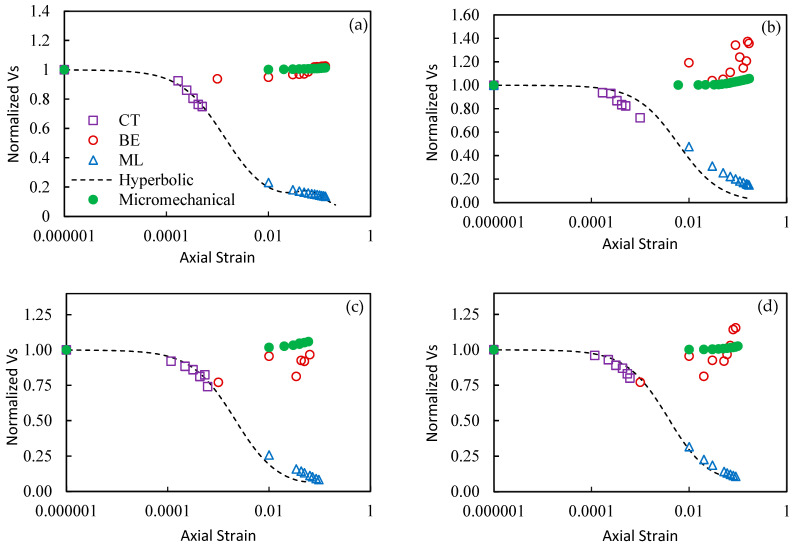
Variation of normalized *Vs* from cyclic triaxial (CT), bender elements (BE), and stress–strain curve of monotonic loading (ML) for (**a**) sample A at 150 kPa, (**b**) sample A at 300 kPa, (**c**) sample B at 150 kPa, and (**d**) sample B at 300 kPa.

**Figure 9 materials-16-00766-f009:**
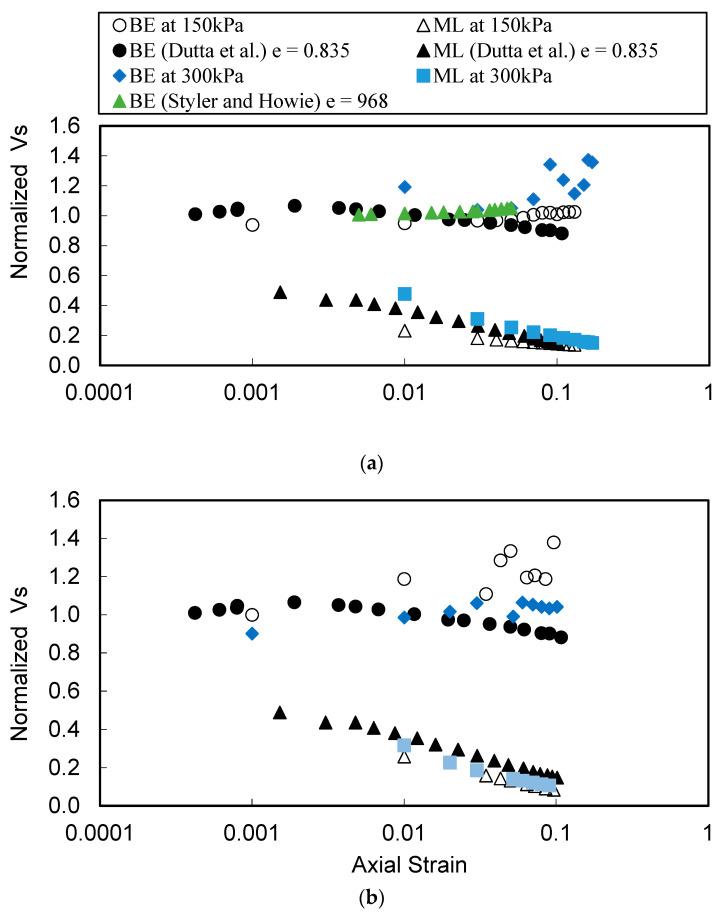
Comparison of results from this study with the results from Dutta et al. [8]; Styler and Howie [9] (**a**) sample A and (**b**) sample B.

**Table 1 materials-16-00766-t001:** Characteristics of RC, CT, BE, and field tests.

Test	Frequencies (Hz) /Strains (e)	Dynamic Properties	Advantages	Disadvantages
RC	20–150 e < 10^−3^	Vs and D	Wide range of frequencies and strains	Many testing biases such as base contributions
CT	0.1–10 10^−4^ < e < 10^−2^	Vs and D	Same equipment can be used for static test	Narrow strain range; requires additional tests
BE	20–150 e < 10^−6^	Vs	Fast and minimal equipment setup	Damping ratio (D) cannot be calculated; narrow strain range
Field tests	20–150 <10^−6^	Vs	Representative *Vs* of the site	Damping ratio (D) cannot be calculated; narrow strain range
Earthquakes	0.1–50 10^−6^ < e < 10^−1^			

**Table 2 materials-16-00766-t002:** Selected correlations between *C_n_* and e for a random assembly of particles.

Expression	Reference
$C_{n}=26.486-10.726\;\left(1+e\right)$	Smith et al. [25]
$C_{n}=\frac{32}{13}\left[7-8\left(\frac{e}{1+e}\right)\right]$	Ouchiyama & Tanaka [26]
$C_{n}=24e^{\left(-2.547\frac{e}{\left(1+e\right)}\right)}-0.373$	Zimmer [27]
$C_{n}=2+11{\left(\frac{1}{1+e}\right)}^{2}$	German et al. [28]
$C_{n}=-2.8+15.1\left(\frac{1}{1+e}\right)$	

**Table 3 materials-16-00766-t003:** Physical properties of the tested specimens and standard penetration test (SPT) values.

Sample Name	Void Ratio (*e*)	C_C_	C_U_	Diameter/Height (cm)	Moisture Content (%)	Dry Density (kg/m^3^)	Material Description	Depth (m)	SPT-N Value
Sample A	0.81	0.82	2.45	7/14.2	22.2	1618	Sand with crystalline gypsum inclusions, very dense and poorly graded	11 to 12	50
Sample B	0.99	1	2.00	7.1/14.0	12	1478	Sand with silt and crystalline gypsum inclusions, very dense and poorly graded	1.5 to 2	21

**Table 4 materials-16-00766-t004:** Summary of the chemical compositions of tested soil samples.

Element	Standard Label	Apparent Voncentration	Weight Fraction (wt%)	Apparent Concentration	Weight Fraction (wt%)
		Sample A		Sample B	
O	SiO_2_	2.52	48.82	2.71	52.74
Na	Albite	0.05	0.78	0.10	1.75
Mg	MgO	0.20	6.18	0.29	6.42
Al	Al_2_O_3_	0.11	2.71	0.08	1.93
Si	SiO_2_	0.87	20.47	0.60	13.74
Cl	NaCl	0.14	1.7	0.13	3.30
Ca	CaSiO_3_	0.28	10.51	0.85	20.12
Fe	Fe	0.16	7.72	0.00	0.00

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
