# Peer review of "Continuous Evaluation of Shear Wave Velocity from Bender Elements during Monotonic Triaxial Loading"

_materials, 2023, doi:10.3390/ma16020766_

Round 1

Reviewer 1 Report

This paper presents the Continuous Evaluation of Shear Wave Velocity from Bender Elements during Monotonic Triaxial Loading. There are some important issues that should be addressed before this paper can be considered for publication, as listed bellow

1.       Authors mentioned that the sands were taken from an urban area in Abu Dhabi. Please highlight and elaborate on the location of the sampling place. Authors also need to present the physical properties of those samples, including the gradation because to show Cu and Cc only is not enough. Why do authors take the sample at depths of 11 to 12 and 1.5 to 2? Since authors present SPT, it would be better to show the soil profile of the site.

2.       In the Background and literature, the authors shortly present the measurement methods including the triaxial test, cyclic triaxial test, and resonance column. However, it is not very comprehensive. I suggest authors to presents the matrix of difference for those measurements, including merit and demerit.

3.       To avoid near-field effects, at least one wavelength was ensured between the transmitter and receiver in this study (Arroyo [44], Lee [45], Khalil et al. [3]; Khalil [22]). The authors should explain about this

4.       Authors explain that the results of this study tend to be different from previous studies because of the heterogeneous nature of the mineral composition of the sand particles. However, this is still questionable and unclear. To me, stating this should be supported by a clear explanation and obvious proof.

5.       Authors mentioned “The stress-strain behavior of tested sands (heterogeneous mineral composition) is markedly different than siliceous sand. Please explain why???

6.       In Figure 7, the trend of the hyperbolic model of Vs and the axial strain shows a good correlation with cyclic triaxial and monotonic cyclic loading instead of BE and micromechanical. The last methods even show very different tendencies. In the text, this phenomenon is still limited discussed. Authors should explain this to enhance the characteristics of BE test results.

7.       To enhance the readership interest, please add the following studies that discussed how is the role of shear wave velocity in a large scale for seismic response analysis.

8.       To me, the experimental test of this study is still lacking and limited. Authors at least are suggested to test again to obtain the correct and strong reasons for some questionable statement that is just presumed by authors.

Author Response

Comment 1.      

Authors mentioned that the sands were taken from an urban area in Abu Dhabi. Please highlight and elaborate on the location of the sampling place. Authors also need to present the physical properties of those samples, including the gradation because to show Cu and Cc only is not enough. Why do authors take the sample at depths of 11 to 12 and 1.5 to 2? Since authors present SPT, it would be better to show the soil profile of the site.

Response

The rough location of the borehole has been added to the paper. The exact name of the project is not added because it will require permission from the authorities and also from the client. The physical properties such as dimensions, density etc have been added to Table 2.

The particle size gradation curves are also added as a new figure (Figure 1). The depths were chosen randomly because the objective of the paper is not to investigate the effect of soil type on stress strain relationship. However, we acknowledge that some statements of the paper shows as if the effect of soil type was investigated. We have modified those statements to indicate that the presumed statement is one of the possible many reasons for difference. The modified statements appear on lines 272, 273, 342, 343, 344, and conclusions. The responses to comments 4 and 5 includes more details.

Comment 2.      

In the Background and literature, the authors shortly present the measurement methods including the triaxial test, cyclic triaxial test, and resonance column. However, it is not very comprehensive. I suggest authors to presents the matrix of difference for those measurements, including merit and demerit.

Response

A new Table 1 indicating the features, merits, and demerits has been added. The text on lines 98 to 102 has been enhanced. The idea was to keep the explanation of test method as short as possible because these tests are ASTM standards where details of the tests can be found.

Comment 3.      

To avoid near-field effects, at least one wavelength was ensured between the transmitter and receiver in this study (Arroyo [44], Lee [45], Khalil et al. [3]; Khalil [22]). The authors should explain about this.

Response

The explanation has been added on lines 328 to 332.

Comment 4.      

Authors explain that the results of this study tend to be different from previous studies because of the heterogeneous nature of the mineral composition of the sand particles. However, this is still questionable and unclear. To me, stating this should be supported by a clear explanation and obvious proof.

Response

The statements have been revised to indicate that the difference in observed behavior could be due to factors other than simple heterogeneous nature of the mineral composition. We agree that the statements like these are superficial and do not contribute to the main objective of the paper. The statement can be removed; however, we decided to keep it. The modified statements appear on lines 272, 273, 342, 343, 344, and conclusions.

Comment 5.      

Authors mentioned “The stress-strain behavior of tested sands (heterogeneous mineral composition) is markedly different than siliceous sand. Please explain why???

Response

Same as comment 4; however, we have now added possible reasons for the difference in behavior of sands in this study compared to literature from monotonic triaxial tests. The modified statements appear on lines 272, 273, 342, 343, 344, and conclusions. We agree that the statement is not the result of parametric and conclusive study because the objective of the paper is rather to explain the behavior of low strain wave propagation (BE tests) through a soil undergoing large strains (monotonic triaxial tests) through micro-mechanical approach. A separate study would be needed to show with high confidence the main reason(s) for the difference in behavior observed through monotonic triaxial tests and their sensitivity.

Comment 6.      

In Figure 7, the trend of the hyperbolic model of Vs and the axial strain shows a good correlation with cyclic triaxial and monotonic cyclic loading instead of BE and micromechanical. The last methods even show very different tendencies. In the text, this phenomenon is still limited discussed. Authors should explain this to enhance the characteristics of BE test results.

Response

The main objective of the paper is to explain with micro mechanical approach the very different tendencies between BE/micro-mechanical and hyperbolic models. Superimposed low strain propagation of waves appears to be unaffected by samples undergoing large strain deformations which is strange behavior, and no clear explanation is present in the literature. This is exactly the issue as pointed out by the reviewer. This paper tries to explain the discrepancy through a micro-mechanical perspective using contact theories.  It appears that micro-mechanical approach is better suited to explain the results of low strain wave propagation. The main discussion of the paper is about this phenomenon; therefore, we are not sure where in the text shall we add more discussions. We have; however, added more discussion about the behavior of cyclic triaxial and monotonic loading with hyperbolic model on lines 358 to 361. We have also added in conclusions a limitation in using contact theories for cohesive soils such as clays.

Comment 7.      

To enhance the readership interest, please add the following studies that discussed how is the role of shear wave velocity in a large scale for seismic response analysis.

Response

The references to the studies do not appear in the text here. We have however added the importance of shear wave velocity in seismic response analysis on lines 25 and 26.

Comment 8.      

To me, the experimental test of this study is still lacking and limited. Authors at least are suggested to test again to obtain the correct and strong reasons for some questionable statement that is just presumed by authors.

Response

We have modified those statements that appear to us to be causing confusion for the reviewer. The statements are modified to illustrate that the suggested reason could be one of many possible reasons. The modified statements appear on lines 272, 273, 342, 343, 344, and conclusions. The presumed statements (mentioned in the reviewer comments above) are not contributing to the main objective. In our opinion the statements can be even deleted without compromising the objective and purpose of the study.

The tests are performed on two identical specimens for each test as stated on lines 62 and 63 of the manuscript. The results were practically identical and repeatable.

Reviewer 2 Report

This study presents the continuous measurement of low-strain Vs with Bender Elements (BE) during monotonic shearing of two sand specimens in a triaxial device. The results confirms that the Vs values from BE measurements are unaffected by increase in axial strains that are induced by separate mechanism. The micro-mechanical model predictions of Vs agree well with the results of this study and with the results of previous studies. It shows that the mean effective stress and increase in interparticle stiffness controls the low-strain stiffness despite global increase in strains during monotonic loading. The paper is generally well written and much of it is well described. I believe that the authors have carried out careful and thorough experimental research and theoretical analysis. This work is relatively interesting, and the structure and language of the manuscript are well prepared. The following comments need to be addressed or considered to improve the technical depth of the manuscript.

1Please specify the source of the sand sample.

2An EDS spectrum named sample B can be displayed

3How to consider the influence of the heterogeneity of sand mineral composition in the stress-strain analysis.

4More CT sample analysis can be added to make the research conclusions more objective.

5In the conclusion part, whether there is something worth further improvement and discussion in this paper, and whether the method in this paper has certain limitations and scope of application, can be added to explain.

Author Response

Comment 1

Please specify the source of the sand sample.

Response

We have added more information about the project location and have added physical properties of the tested samples in a table xxx. We have also added the gradation curves for the sand samples in new figure 1. The exact name of the project is not added because it will require permission from the authorities and also from the client.

Comment 2

An EDS spectrum named sample B can be displayed

Response

The figure has been added as Figure 3b

Comment 3

How to consider the influence of the heterogeneity of sand mineral composition in the stress-strain analysis.

Response

The statement by the authors is very superficial and is not the main objective of the paper therefore we have modified the occurrences of this claim to show that this is one of the possible reasons. In our opinion deletion of this statement will also not affect the main contributions. The modified statements appear on lines 272, 273, 342, 343, 344, and conclusions.

Comment 4

More CT sample analysis can be added to make the research conclusions more objective.

Response

More analysis and discussions have been added on lines 358 to 361.

Comment 5

In the conclusion part, whether there is something worth further improvement and discussion in this paper, and whether the method in this paper has certain limitations and scope of application, can be added to explain.

Response

The conclusions have been amended accordingly.

Reviewer 3 Report

Shear wave velocity (Vs) under axial strain at monotonous triaxial compression was obtained by bender element, cyclic triaxial tests and stress-strain curves. The results were compared with the known data and showed good correlation for low-strain Vs and the significant difference in stress-strain behavior of tested heterogeneous sands siliceous sand. Variation of Vs obtained from bender element tests is well approximated by contact theory. Vs calculated from the stress-strain curves and cyclic triaxial tests are mostly well approximated by hyperbolic model.

Test results of the systems and structural materials under complex contact and non-contact loading together with volumetric damage models could be referred to in the paper:

(i) a method of experimental study of friction in a active system, (ii) state of volumetric damage of tribo-fatigue system, (iii)  spatial stress-strain state of tribofatigue system in roll-shaft contact zone, (iv) modeling of the damaged state by the finite-element method on simultaneous action of contact and noncontact loads, (v) tribo-fatigue behavior of austempered ductile iron monica as new structural material for rail-wheel system, (vi) research on tensile behaviour of new structural material monica, (vii) measurement and real time analysis of local damage in wear-and-fatigue tests

Lines 27 and 30 have repeating phrases “have limitations and testing biases”.

f(mu) in line 164, SPT-N in table 2, Tx in fig. 5 should be defined. Other entities should also be checked for definition when introduced.

The paper “Continuous Evaluation of Shear Wave Velocity from Bender Elements during Monotonic Triaxial Loading” could be published in Materials after minor revision.

Author Response

Comment 1

Test results of the systems and structural materials under complex contact and non-contact loading together with volumetric damage models could be referred to in the paper:

(i) a method of experimental study of friction in a active system, (ii) state of volumetric damage of tribo-fatigue system, (iii)  spatial stress-strain state of tribofatigue system in roll-shaft contact zone, (iv) modeling of the damaged state by the finite-element method on simultaneous action of contact and noncontact loads, (v) tribo-fatigue behavior of austempered ductile iron monica as new structural material for rail-wheel system, (vi) research on tensile behaviour of new structural material monica, (vii) measurement and real time analysis of local damage in wear-and-fatigue tests

Response

Added on lines 417 to 422

Comment 2

Lines 27 and 30 have repeating phrases “have limitations and testing biases”.

Response

Corrected.

Comment 3

f(mu) in line 164, SPT-N in table 2, Tx in fig. 5 should be defined. Other entities should also be checked for definition when introduced.

Response

Corrected.

Round 2

Reviewer 1 Report

Accept